# An Empirical Evaluation of a Novel Ensemble Deep Neural Network Model and Explainable AI for Accurate Segmentation and Classification of Ovarian Tumors Using CT Images

**DOI:** 10.3390/diagnostics14050543

**Published:** 2024-03-04

**Authors:** Ashwini Kodipalli, Steven L. Fernandes, Santosh Dasar

**Affiliations:** 1Department of Artificial Intelligence and Data Science, Global Academy of Technology, Bangalore 560098, India; 2Department of Computer Science, Design, Journalism, Creighton University, Omaha, NE 68178, USA; 3Department of Radiology, SDM College of Medical Sciences & Hospital, Shri Dharmasthala Manjunatheshwara University, Dharwad 580009, India; dr.santoshdasar@sdmuniversity.edu.in

**Keywords:** U-Net, transformers, multi-model ensemble, ovarian cancer, benign, malignant, computer-aided diagnosis, LIME, SHAP

## Abstract

Ovarian cancer is one of the leading causes of death worldwide among the female population. Early diagnosis is crucial for patient treatment. In this work, our main objective is to accurately detect and classify ovarian cancer. To achieve this, two datasets are considered: CT scan images of patients with cancer and those without, and biomarker (clinical parameters) data from all patients. We propose an ensemble deep neural network model and an ensemble machine learning model for the automatic binary classification of ovarian CT scan images and biomarker data. The proposed model incorporates four convolutional neural network models: VGG16, ResNet 152, Inception V3, and DenseNet 101, with transformers applied for feature extraction. These extracted features are fed into our proposed ensemble multi-layer perceptron model for classification. Preprocessing and CNN tuning techniques such as hyperparameter optimization, data augmentation, and fine-tuning are utilized during model training. Our ensemble model outperforms single classifiers and machine learning algorithms, achieving a mean accuracy of 98.96%, a precision of 97.44%, and an F1-score of 98.7%. We compared these results with those obtained using features extracted by the UNet model, followed by classification with our ensemble model. The transformer demonstrated superior performance in feature extraction over the UNet, with a mean Dice score and mean Jaccard score of 0.98 and 0.97, respectively, and standard deviations of 0.04 and 0.06 for benign tumors and 0.99 and 0.98 with standard deviations of 0.01 for malignant tumors. For the biomarker data, the combination of five machine learning models—KNN, logistic regression, SVM, decision tree, and random forest—resulted in an improved accuracy of 92.8% compared to single classifiers.

## 1. Introduction

One in every eight women worldwide is affected by ovarian cancer, as per the World Ovarian Cancer Coalition. As per the reports, one in every eight women affected is the current incident rate. Ovarian cancer remains a formidable adversary in oncology, posing significant challenges to early detection and effective treatment; as one of the most gynecological malignancies, it often remains asymptomatic in its early stages, leading to diagnoses in advanced, less treatable phases. The crucial role of early detection in improving ovarian cancer prognosis cannot be overstated. Accurate and timely diagnosis is the cornerstone of effective treatment strategies, ultimately influencing patient survival rates and quality of life [1]. This cancer is diagnosed by detecting malignant cells in ovarian tissue. In the pursuit of more precise and reliable diagnostic tools, medical imaging, particularly computed tomography (CT) scans, has emerged as a vital component due to its ability to provide detailed anatomical information and assist in tumor characterization. However, accurately interpreting CT images for ovarian cancer diagnosis remains complex and challenging, often reliant on the expertise of radiologists, introducing inherent subjectivity and potential variability in results [2].

Advancements in medical image processing and computational techniques, including computer-aided mechanisms, are used to achieve improved results compared to manual radiologist findings [3]. In the deep learning domain, this typically involves extracting features using a convolutional neural network (CNN) and classifying them using a fully connected network. Deep learning is widely applied in medical imaging, as prior expertise in the related field is not required.

In medical image processing, convolutional neural networks (CNNs) have been extensively used and have achieved significant results in tasks like image classification and segmentation [4]. CNNs are designed to capture spatial relationships such as image classification, segmentation, and object detection. However, transformers have recently gained popularity in medical image analysis, showing promising results in various tasks. Transformers’ main advantage over CNNs is their ability to handle long-range dependencies and relationships between pixels in an image. In a medical image, features in different regions can be related and significantly impact diagnosis or treatment. Transformers, with their self-attention mechanism, can effectively capture these relationships and dependencies, leading to improved performance in tasks like lesion classification or segmentation. This self-attention mechanism allows for parallel processing, making transformers faster than CNNs and UNet. Another advantage of transformers is their ability to be trained on large datasets, enabling them to learn more complex representations of medical images. However, transformers lack performance when the size of the dataset is limited. This is particularly crucial in medical imaging, where large datasets are often not available [5].

The main contributions of this research work are summarized below:Implemented transformer models for semantic segmentation in ovarian tumor detection and compared the results with the UNet model. Transformer models outperformed the UNet in the segmentation.Conducted a thorough evaluation of segmentation models, comparing the transformer-based approach with the widely recognized UNet model. This assessment involved the application of metrics such as the Dice score and the Jaccard score.Developed a four-stage deep learning ensemble (comprising VGG16, ResNet 152, Inception V3, and DenseNet 101) and a five-stage machine learning ensemble for classifying ovarian tumors.Established the superiority of the ensemble models by demonstrating enhanced classification accuracy in ovarian tumor detection compared to individual classifiers.Implemented explainable AI methodologies, including SHAP and LIME, to enhance the interpretability of the model’s predictions. This approach ensures a transparent understanding of the key features influencing classification outcomes.

This research paper is organized as follows: Section 2 provides an overview of related work in the fields of ovarian tumor classification, deep neural networks (DNNs), ensemble methods, UNet, and Transformers. Section 3 details our proposed methodology, including the architecture of the ensemble DNN and the integration of segmentation techniques. Section 4 presents the experimental setup, results, and performance evaluation. In Section 5, we discuss the implications of our findings. We conclude by highlighting the potential of our approach to revolutionize ovarian tumor diagnosis in Section 6. Section 7 provides the discussion. 

## 2. Related Work

Maithra et al. [6] investigated the effect of scale on transfer learning, finding that larger transformer models develop significantly stronger intermediate representations through larger pretraining datasets. Additionally, they analyzed the internal representation structure of transformer models and CNNs in image classification benchmarks, noting substantial differences between the two architectures, such as transformers having more uniform representations across all layers. Alexey et al. [7] present a large-scale study comparing transformer and CNN models by their performance in image classification tasks. The authors demonstrate that transformer models can achieve excellent performance on numerous benchmark datasets and are computationally efficient and easier to train than CNNs. Han et al. [8] compare the performance of transformer and CNN models in generative tasks, such as image synthesis. They show that transformer models can generate high-quality images and outperform CNNs in both quality and sample diversity. The proposed Self-Attention Generative Adversarial Network (SAGAN) provided an inception score of 52.52 and reduced the inception distance from 27.62 to 18.65.

Gao et al. [9] propose a model combining CNNs and transformers to efficiently extract low-level features of images and establish long-range dependencies between modalities. Their research asserts that transformers significantly contribute to multi-modal image analysis compared to CNNs, achieving an improvement of 10.1% in average accuracy compared to the state-of-the-art CNN models. Kelei et al. [10] found that CNNs neglect long-term dependencies within images, such as the nonlocal correlation of objects. In contrast, transformer models overcome such hurdles by formulating image classification as a sequence prediction task for image patch sequences, thereby capturing long-term dependencies within the input image. Fahad et al. [11], in their comprehensive review of transformers in medical image analysis (detection, classification, segmentation, reconstruction, etc.), indicate that transformers, compared to pure CNNs, provide a clearer, localized picture of attention in most medical images and mention the rapid growth of transformers in this field. The proposed method achieved a sensitivity of 91.5% and 82.2% sensitivity.

Emebob et al. [12] experimented with transformers and CNNs using ImageNet pretrained weights, typically the most popular method to improve deep learning model performance. Their results suggest that transformers benefit more from transfer learning, resulting in better-trained models for image analysis. Chang et al. [13] proposed combining transformers and CNNs to overcome the lack of long-range dependencies inherent in CNNs. This combination yields better outcomes with electroencephalogram (EEG) results compared to earlier proposed CNN and DCNN-based models. Hu et al. [14] propose a transformer-based model for medical image segmentation. Their work highlights the advantages of transformers in designing a Swin-Unet, a transformer approach to the segmentation model. After testing multiple U-net-based models and the transformer-based model, they summarize the strength of transformer-based models in image segmentation. The proposed UNETR method obtained the Dice score of 0.964 on the CT spleen and 0.789 on the whole tumor of the brain. Ali et al. [15] proposed a model that includes encoders to learn sequence images of the input volume and successfully captures the global multiscale information, demonstrating the model’s ability to capture global contextual representation at multiple scales. The experimental results show that TransClaw U-Net is better than the other network models for segmentation. Yao et al. [16] demonstrate the ability of a transformer-based model to extract the global context, a crucial factor in medical examination. Their study explores the strength of transformers in medical image analysis, particularly in detailed segmentation performance.

Christos et al. [17], in their paper “Is it time to replace CNNs with transformers for medical images?”, discuss several reasons for transformers’ ability to match the efficiency of CNN models in medical image analysis. They also highlight the role of transfer learning in yielding better results and mention dataset size as a key factor in measuring the performance of transformer-based models. Rene et al. [18] highlight the strength of transformers in semantic segmentation, considering various parameters that can enhance the ability of a transformer in the segmentation process. Zhang et al. [19] show that low-level spatial details in medical image segmentation can be efficiently captured with a combination of transformers and other neural networks. Their extensive experiments reflect the ability of transformers in both 2D and 3D image processing. A study by Guoping et al. [20] demonstrates how a U-Net-based encoding block can increase efficiency and reduce complexity in a transformer’s computation. They propose a viable technique for medical image segmentation, primarily using transformer features on various benchmarks such as ACDC and Synapse. Often, models require large datasets for better training. The proposed transformer-based UNet provided an accuracy of 78.53% with a segmentation speed of 85 frames per second. The proposed model outperformed the CNN model. Jeya Maria Jose et al. [21] propose a local-global training strategy (LoGo) that operates on whole images and patches to learn global and local features, overcoming the shortcomings of training with small datasets.

Miranda et al. [22] discuss various U-Net-based models for medical image segmentation, outlining their shortcomings and highlighting the role of transformers in efficient segmentation. Feiniu et al. [23] list the advantages of using transformers in medical image segmentation, emphasizing their significance in the domain of medical imaging analysis. Zhuangzhuang et al. [24] propose a method to reduce the computational complexity of transformers and compare the results with other state-of-the-art models, showing significant complexity reduction while maintaining stable performance with popular databases. Hongyu et al. [25] compare a CNN and transformer-based hybrid model, demonstrating better feature extraction compared to pure CNN or U-Net models. Their experiments yield significant results, surpassing U-Net-based models on similar datasets. Shen et al. [26] propose combining a transformer with the U-Net model for improved accuracy, also discussing the shortcomings of U-Net in feature extraction and the robustness of transformers in image analysis.

## 3. Methodology

### 3.1. Segmentation

#### 3.1.1. Segmentation Using Transformer Model

Semantic segmentation has a wide range of applications in computer vision and primarily involves assigning each image pixel to a class or category label. Fully convolutional networks have been predominantly used for semantic segmentation tasks. However, in recent years, transformers that rely on the self-attention mechanism have proven to be more efficient in segmentation tasks.

The transformer follows an encoder–decoder architecture. Introduced in the paper ‘Attention Is All You Need’, [27] transformers are described as a Seq2Seq (sequence-to-sequence) architecture. The encoder maps the input sequence to a higher dimensional space, which is then fed to the decoder to produce the output sequence. In the context of semantic segmentation, the transformer maps a sequence of patch embeddings to pixel-level class annotations. Figure 1 illustrates the architecture of the transformer model for segmentation. The details of the transformer model are as follows: the model used is Seg-L, the backbone architecture used is ViT-L with 24 layers, 1024 token size, and having 16 heads with the total parameters 307 M. The recently introduced vision transformer demonstrates an architecture that is free of convolutions, purely processing images as a sequence of patch tokens. Using the encoder–decoder transformer architecture, we have segmented input images of benign and malignant tumors. The performance on benign images is better than that on malignant images due to the clear, well-defined boundaries of benign tumors in comparison to malignant tumors. Malignant tumors, often lacking well-defined boundaries and shapes, are segmented with lower accuracy. 

#### 3.1.2. Segmentation Using U-Net Model

U-net is one of the popular semantic image segmentation models introduced by Ronneberger et al. [28]. The model comprises a U-shaped architecture that applies a downsampling (encoding) and upsampling (decoding). The architecture of the U-net is displayed in Figure 2.

The model identifies the objects present in the image through the encoding stages, also known as the contracting network. Each time the image passes through a layer of encoding, the number of pixels is reduced by half. This reduction is a crucial step in semantic segmentation, the process of assigning a class to each pixel in the image. The decoder, also referred to as the expansion network, processes the feature map received from the lower layers to produce a segmentation mask. The skip connections, which are key to the U-Net’s efficiency, combine the feature map generated at each stage of encoding with the corresponding stages of decoding. These connections, indicated by the grey arrows in Figure 2, create a segmentation map from the contextual features learned throughout the encoding cycles. They also help the model maintain minimal changes in the image’s intensity. The encoder and decoder, together, implement the tasks of classification and localization, respectively. The bottleneck layer of the architecture consists of two convolutional layers followed by a ReLU activation layer. This layer is responsible for generating the final feature map that feeds into the first layer of the decoder.

### 3.2. Classification Using Deep Learning for CT Scan Images

#### 3.2.1. Proposed Network Architecture

The use of computed tomography (CT) scanned images for detecting ovarian cancer using deep learning is not widespread. This study is one of the few that utilizes ensemble deep learning to achieve its objective. An ensemble deep learning model, built as a combination of multiple CNN models, aims to achieve better accuracy. The number of layers and the extent of learning enable the model to effectively extract the required features from the input image. Figure 3 illustrates the approach of the proposed task, which is divided mainly into these steps: data collection, preprocessing and dataset preparation, feature extraction, segmentation, and classification.

In the first step, CT scan images of several patients are collected with their consent. Preprocessing involves removing sensitive or irrelevant information from the images. Data augmentation is also performed to increase the dataset size. The proposed four-path ensemble architecture is then fed with this data as input, meaning the four CNN models involved receive this data and perform feature extraction separately. The reason why the segmentation was performed 1st and then classification was that the raw images contained a large amount of information, including irrelevant details, not the lesion of interest. Therefore, segmentation was applied 1st so that the model could focus only on the lesion of interest, thus reducing the complexity of the classification model. Instead of considering every pixel for classification, features such as shape can be extracted from the segmentation model, thus providing more discriminative information for classification. This benefits the feature extraction for the classification. Since the segmented region is given to the classifier, the classification model can be tailored to the characteristics of those regions, leading to improved accuracy—computational efficiency as we are processing only the segmented images rather than the entire image. The resulting vectors are combined to form a multiview feature vector, which is then sent to a multi-layer perceptron architecture for classifying the cases into two categories: benign and malignant. Evaluation metrics such as accuracy, precision, recall, and F-score are used to assess the proposed model’s performance.

#### 3.2.2. Feature Extraction Using Transfer Learning

Feature extraction plays a crucial role in the classification of histopathological images using deep learning due to their high visual complexity. This aspect directly impacts the performance of the CNN model in use. Privacy concerns related to medical data limit the size of the dataset that can be acquired. To enhance performance, alongside data augmentation, transfer learning proves to be a beneficial method for better feature extraction. Models trained to extract general features from one dataset can be effectively applied in different scenarios involving other objects of interest.

#### 3.2.3. Four-Path Ensemble Architecture for Ovarian Cancer Classification

The proposed system is a four-path ensemble architecture for ovarian cancer detection, utilizing four popular deep-learning classifiers: VGG16, ResNet 152, Inception V3, and DenseNet 101. These models were selected for their high accuracy from a pool of commonly used CNN models, including VGG16, VGG19, ResNet 152, Inception ResNet V1, Inception ResNet V2, EfficientNet B1, and DenseNet 101. Figure 4 illustrates the architecture of the proposed system and the detailed implementation algorithm is given in Algorithm 1. The input data undergoes preprocessing such as intensity normalization, image resizing, image enhancement using Gaussian filters, anatomical normalization, and data augmentation before being fed into all four CNN models. Each model, depending on its algorithm, performs feature extraction and learning. The last fully connected layers of these models are then combined to form a unified feature vector, aiding the four-path ensemble deep learning model in classifying the instances as benign or malignant. Karen Simonyan and Andrew Zisserman named VGGNet after the Visual Geometry Group at the University of Oxford in 2014. It was one of the top performers in the ImageNet Large Scale Visual Recognition Challenge (ILSVRC). VGGNet’s architecture consists of 3 × 3 convolutional layers stacked on top of each other, alternated with a max pooling layer, followed by two fully connected layers with 4096 nodes each, and a SoftMax classifier at the end. Residual networks, or ResNets, were developed to address the vanishing/exploding gradient problem associated with adding too many layers to a deep neural network. They use skip connections in the residual blocks, which are stacked to form ResNets. The ResNet 152 model, which has 152 layers, is notable for having fewer parameters than the VGG19 model and for winning the ILSVRC ImageNet 2015 challenge. Building on the success of ResNets, Inception-ResNet models, such as Inception-ResNet V1 and V2, were developed. These models incorporate inception blocks, which are computationally less expensive, with residual connections replacing the pooling operations in the reduction blocks. Additionally, batch normalization is not used after summations, and filter expansion occurs after each inception block. The computational costs of Inception-ResNet V1 and V2 are similar to those of Inception V3 and V4, respectively. DenseNet, proposed by Huang et al. in 2016, features a ‘dense block’ in its architecture, wherein each convolutional layer is directly connected to all subsequent layers. DenseNet, short for densely connected convolutional networks, has a complex interconnected structure with very short connections between input and output layers throughout the network, which helps in mitigating the vanishing gradient problem. The configuration of each of the variants of CNN is described in Table 1.

#### 3.2.4. Algorithm for Segmentation Using UNet and Transformers 

**Algorithm 1:** Algorithm for Segmentation using UNet and Transformers**Input:** X as the input image, $c_{i}$ as the output of the ith convolutional layer in the UNet model, $f_{i}$ as the ith feature map, $p_{i}$ as the ith pooling layer, $u_{i}$ as the ith up-convolutional (transposed convolutional) layer in the U-Net model, $t_{i}$ as the output of the transformer model, $m_{i}$ as the multi-head attention mechanism in the transformer model, $d_{i}$ as the ith dense layer in the transformer model.
**Output:** Segmented Lesion
**1. for** i in range(N) **do:**            E = sigma(Conv(E-1))           D = sigma(Conv(D-1)            S = E + D            L_bce = sum(y_i × log(O_i) + (1 − y_i) × log(1 − O_i) for i in range(N))            L_dice = (2 × TP)/((TP + FP) + (TP + FN))            L_jaccard = TP/(TP + FP + FN)**end for**# Transformer-style segmentation with MultiHeadAttention, Position-wise Feed-forward Network, and Normalization**2. for** i in range(N) **do:**                      E = MultiHeadAttention_Layer(x) + x                      D = MultiHeadAttention(Y) + Y + MultiHeadAttention(Encoder_Output)            FFN = ReLU(Conv1D(Z, W_1 + b_1)) @ W_2 + b_2**end for**

### 3.3. Classification Using Ensemble Machine Learning Model for Biomarker Dataset 

The classification process for the biomarker dataset involves meticulous preprocessing of the tabulated biomarker (clinical parameters) data. This preprocessing includes feature selection, handling missing values, and transforming the dataset for a more concise representation. The goal is to ensure that each feature encapsulates analogous examples. Following these preprocessing steps, the dataset undergoes classification using machine learning classifiers, namely K-nearest neighbors (KNN), logistic regression, support vector machine (SVM), random forest, and decision tree.

Ensemble learning is employed to harness the diverse performances of these classifiers, aiming to achieve an optimal outcome. The key to superior performance lies in careful hyperparameter tuning. To elucidate and interpret this optimal performance, explainable AI methods such as LIME (local interpretable model-agnostic explanations) and SHAP (SHapley Additive exPlanations) are utilized. These methods provide transparency in understanding the influential factors contributing to the outcomes of the ensemble machine learning model (Figure 5).

#### Algorithm 2 for Classification Using Ensemble Machine Learning Models and Interpretation Using LIME and SHAP


**Algorithm 2:** Ensemble Machine Learning Model Step 1: Data PreparationStep 2: Base Model Training            base_models = {                   ‘Logistic Regression’: LogisticRegression(),                   ‘KNN’: KNeighborsClassifier(),                   ‘SVM’: SVC(),                   ‘Decision Tree’: DecisionTreeClassifier(),                   ‘Random Forest’: RandomForestClassifier()            }base_model_predictions = {}            for model_name, model in base_models.items():                   model.fit(X_train, y_train)                   base_model_predictions[model_name] = model.predict(X_test)Step 3: Base Model Predictions            base_model_predictions_array = np.array(list(base_model_predictions.values())).TStep 4: Meta-Model Training            meta_model = Ensemble()           meta_model.fit(base_model_predictions_array, y_test)Step 5: Final Prediction            final_predictions_array = np.array(list(base_model_predictions.values())).T            stacked_predictions = meta_model.predict(final_predictions_array)Step 6: Evaluation            accuracy = accuracy_score(y_test, stacked_predictions)            print(f”Stacking Ensemble Model Accuracy: {accuracy}”)Step 7: Explainability with LIME            def lime_explanation(model, instance, features):Step 8: Explainability with SHAP            explainer = shap.Explainer(meta_model)            shap_values = explainer.shap_values(X_test)


## 4. Experiments

### 4.1. Dataset Description

A total of 349 anonymous patient CT scan images were collected from SDM Medical College and Science, Dharwad. The dataset consists of 540 benign and 487 malignant images. Each image has a resolution of 512 × 512 pixels. The dataset includes axial, coronal, and sagittal views to aid in evaluating the extent of the disease. In contrast, the biomarker dataset comprises 349 entries with 50 features, including clinical biomarkers, blood parameters, cancer antigen levels, liver enzymes, and hematological indices. This dataset is split into training and testing sets with an 80:20 ratio, allocating approximately 280 instances for model training and 69 instances for testing.

### 4.2. Data Preparation and Preprocessing Technique

To optimize the training process, data augmentation and image normalization were adopted. 

#### 4.2.1. Data Augmentation

Deep learning algorithms require a substantial amount of data for the training process to effectively understand the patterns within the data. Due to the limited availability of data and to prevent overfitting during the training process, data augmentation is carried out. This process generates additional data from the existing dataset. Various data augmentation techniques, such as horizontal flipping, vertical flipping, contrast enhancement, and adjustments with a zoom and shear range of 0.2 and a rotational range of 90°, are applied to the dataset to create more training samples, thus increasing the samples which will improve the performance of the model.

#### 4.2.2. Image Normalization

Intensity normalization is applied to attain the same range of values for each input image before feeding into the CNN model. This process will help in speeding up the convergence of the model. The input images are normalized using min–max normalization to the intensity range between 0 and 1.

### 4.3. Experimental Settings

The following were the experimental settings at the time of training and execution of the modified U-net model, as shown in Table 2.

The images are passed through bicubic interpolation for the resizing. A fully connected final layer with a ReLU activation function followed by a dropout layer with a probability of 0.5 is used. The intention behind this dropout layer is to avoid overfitting. The Adam optimizer is used in this experiment with the beta 1 and beta 2 parameters with the values 0.6 and 0.8. A 0.0001 learning rate is set for the model. Two classes, namely benign and malignant, are the possible output classifications. All pretrained CNN models are fine-tuned separately. The Keras package in Python is the core behind the implementation of the architecture.

### 4.4. Evaluation Metrics

The performance of the proposed model is evaluated based on accuracy, precision, recall, and F1 score. Mathematically, the metrics are expressed as follows:$$Accuracy=\frac{TP+TN}{TP+TN+FP+FN}\times 100$$
$$Precision=\frac{TP}{TP+FP}\times 100$$
$$Recall=\frac{TP}{TP+FN}\times 100$$
$$F1\;score=\frac{2\times Precision\times Recall}{Precision+Recall}$$

## 5. Results and Discussion

### 5.1. Segmentation

The segmentation results obtained using the transformer model and U-Net model, respectively, are compared in the subsections below with respect to the performance metrics defined.

#### 5.1.1. Performance Metrics

The performance metrics applied to evaluate the segmentation results are listed below:

Dice score: This metric measures the similarity between the two images—the ground truth image and the segmented image. The formula for the Dice score is given below:$$Dice=\frac{2*|S\cap G|}{\left|S\right|+|G|}$$
where *S* indicates the segmented region predicted by the model and *G* indicates the ground truth segmented region. |.| indicates the cardinality of the set. The *Dice* score ranges between 0 and 1, and the closer the score is to 1, the better the segmentation results.

Jaccard score: This metric calculates the area of overlap between the segmented and the ground truth. The formula for the Jaccard score is given below:$$Jaccard=\frac{Dice}{2-Dice}$$

The Jaccard score ranges between 0 and 1. The higher the value, the better the segmentation results.

#### 5.1.2. Comparison of UNet and Transformers 

Figure 6 and Figure 7 and Table 3 depict the segmentation results of UNet and transformers on the open dataset. During testing, the data could randomly be picked from either the training dataset or the validation/test dataset, making the model well-trained. This setup ensures that the segmentation model becomes more robust and performs better in real-world scenarios. The results illustrate that the transformer model outperformed UNet, with a Dice score of 0.98 and a Jaccard score of 0.97 for benign images and a Dice score of 0.99 and a Jaccard score of 0.98 for malignant images.

The segmentation model UNet was trained for 5000 and transformers for 6000 epochs, respectively. From Figure 8, it can be noted that the models are well trained, and no saturation was observed.

### 5.2. Classification of Ovarian Tumors Using Deep Learning

The current work focuses on developing a four-model ensemble deep neural network to classify ovarian tumors. The performance of the classifiers was individually evaluated using pretrained models such as VGG16, DenseNet 101, Inception V3, and ResNet 152, with modifications made to their final layers. The mean accuracy was calculated by running each classifier for five iterations and recording their accuracy. These values are tabulated and presented in Table 4. The table clearly indicates that DenseNet 101 outperforms the other classifiers, achieving a mean accuracy of 97.7%. DenseNet utilizes a compound scaling method that optimally balances the depth, width, and resolution of the model while also using computational resources more effectively to process both low-level and high-level features. Often, essential information in CT scan images is distributed across different scales, and this technique successfully captures these significant details more effectively than the other models. 

Individual classifiers are merged into an ensemble to reduce the variance of each. When several models are trained on the same data, they may exhibit different errors due to their varied characteristics. Merging their outputs helps mitigate the impact of these individual errors, thereby producing more stable and reliable predictions. This approach effectively captures different aspects of the data distribution. Different classifiers may excel in different regions of the feature space within the same dataset or sometimes on different subsets of the data. The diverse perspectives observed in each model contribute to a more robust ensemble model, which typically demonstrates better accuracy and is likely to perform better on unseen or test data. The performance of the ensemble model is presented in Table 5 and Figure 9. The color line in Figure 9 indicates the mean value. 

In this study, we developed and evaluated a newly proposed ensemble deep convolutional neural network (CNN) model for classifying CT scan tumors as benign or malignant. The proposed ensemble model exhibited excellent performance, achieving a mean accuracy of 98.96%. A figure this high signifies that the model accurately predicts a significant portion of the test data. The model also achieved a precision of 97.44%, highlighting its ability to precisely classify positive instances and reduce false positives. Furthermore, the F1 score, which balances precision and recall, reached 98.7%, emphasizing the robustness and effectiveness of the CNN model in correctly identifying both true positive and true negative instances.

### 5.3. Classification Using Machine Learning

As previously noted, the study included a comparison between outcomes generated by deep learning models for CT image data and those derived from the biomarker dataset. Considering the tabular structure of the gathered data, various machine learning models such as logistic regression, KNN, SVM, decision tree, random forest, and boosting methods were employed. The performance metrics of these individual classifiers are detailed in Table 6. The performance of SVM and the random forest was further enhanced with hyperparameter tuning techniques such as manual hyperparameter, Randomized searchCV, and Grid Search CV. The results of the hyperparameter tuning are tabulated in Table 7. Subsequently, an ensemble model was created by amalgamating the results from these individual classifiers to enhance overall accuracy. The summarized results of the ensemble model are presented in Table 8.

Combining the strengths of individual machine learning classifiers, the ensemble model achieved an enhanced accuracy of 92.85%. Precision and F1 Score metrics, crucial for evaluating classification models, were notably high at 97% and 94%, respectively.

### 5.4. Explainer Models

The research employed LIME and SHAP explainer models to interpret and provide insights into the results obtained from machine learning models applied to the biomarkers dataset.

#### 5.4.1. LIME Explainer

Utilizing the Local Interpretable Model-Agnostic Explanation (LIME) technique to interpret results from machine learning models, our investigation into the biomarkers dataset has unveiled a compelling revelation. The standout features indicative of a positive condition, namely the presence of an ovarian tumor, are as follows: HE4, with a noteworthy value of 42.17; CA125 at a striking value of 17.46; AFP registering at 1.25; AST displaying a significant 25.00; and CL making its mark at 104.50, as shown in Figure 10a,b.

The interpretation of the LIME results is as follows for Figure 10a sample 1: For the test sample shown in Figure 10a, the model predicted the sample as malignant, indicating ‘1’ as shown on the left side of the figures. The features that contributed to predicting the samples as malignant are indicated in orange. Since three features contribute towards the positive side, the resultant prediction is malignant. The weights assigned by the model for the features HE4, CA125, and AFP are 0.22, 0.11, and 0.11, respectively. The feature values shown on the right side of the image indicate the actual values of the features present in the dataset. The model considered these three features because the actual values present in the dataset for the samples are less than the threshold values. For example, the actual value of HE4 for the given sample is 42.17, while the model value is 42.53. Since the actual value is less than 42.53, the model considers HE4 as a positive contributing feature. AST and CL contribute towards the negative side; these two features have not contributed to the prediction of malignancy for the given test sample. For sample 2, as shown in Figure 10b, the model was predicted as benign for the test sample passed, indicating ‘0′ as shown on the left side of Figure 10b. 

#### 5.4.2. SHAP Explainer

SHAP values also play a pivotal role in elucidating the significance of individual features when predicting a specific instance. These values effectively allocate the prediction value across input features, thereby providing valuable insights into the specific importance of each feature in the prediction process. A careful examination of the graph below reveals 49 features and their corresponding importance values, where we can observe that HE4 is one of the top features. Delving further into the intricacies, one can uncover the nuanced relationships and dependencies among these features. This analytical approach allows for a comprehensive understanding of the intricate dynamics involved in making predictions, as shown in Figure 11.

The interpretation of the SHAP explainer is as follows: Figure 11a displays the top 20 features based on their weights as considered by the models. Since SHAP serves as the global interpreter, Figure 11a showcases the features of the overall test samples. From the test dataset, we randomly selected a subset of 50 samples and interpreted the predictions on these 50 samples. It is observed from Figure 11b that the features considered of high importance on the overall test samples remain important even on the subset data. Figure 11c elucidates the contribution of the HE4 feature to each of the test samples, as depicted in 11c. The contribution of HE4 to the sample ID 196 from the test dataset is 0.1937. All the points above the horizontal line in yellow and orange represent the samples for which the impact of HE4 is positive, leading to malignant samples, while all the points below the line in blue and gray represent those samples for which the impact of HE4 is negative, leading to benign results. Figure 11d demonstrates the interpretation for the single test sample with the index ID 219. Since the number of features contributing towards the positive side, indicated by the yellow bar lines, is greater than the features contributing towards the negative side, indicated in gray, the test sample 219 is identified as malignant, with the major features contributing being HE4, CA72-4, CA125, and ALB.

## 6. Conclusions and Future Scope

In conclusion, this research introduces an ensemble-based deep-learning approach for the accurate diagnosis of ovarian cancer. By leveraging the transformer for feature extraction and combining the strengths of prominent CNN models such as VGG19, ResNet 152, Inception-ResNet V4, and DenseNet 169, the ensemble model demonstrates superior generalization performance with an accuracy of 98.96%, compared to individual classifiers. The experimental findings not only showcase the model’s superiority over single classifiers but also its ability to surpass state-of-the-art machine learning algorithms across all test set samples. Remarkably, transformers exhibit enhanced performance, even excelling in the detection of small malignant tumors compared to UNet. This underscores the effectiveness of the proposed deep learning multiensemble model in elevating prediction performance beyond the capabilities of base architectures.

In addition to the deep learning approach, this research incorporates an ensemble machine learning model, leveraging the combined results of various classifiers to achieve an enhanced classification accuracy of 92.85%. Furthermore, the application of explainable AI (XAI) methodologies, such as SHAP and LIME, proves invaluable in identifying and interpreting the key features influencing classification outcomes. This not only ensures a transparent understanding of the model’s predictions but also offers a practical advantage by potentially reducing the need for extensive blood tests. XAI aids in pinpointing crucial features, streamlining the diagnostic process, and providing valuable insights for informed decision-making in ovarian cancer detection.

Looking ahead, the research suggests promising directions for the ongoing enhancement and application of the developed ensemble-based deep learning approach for ovarian cancer detection. The integration of multi-modal data, such as genetic information or additional clinical parameters, could further enrich the model’s understanding and diagnostic capabilities. Future efforts may focus on real-time implementation in clinical settings, facilitating swift and accurate diagnoses for timely interventions. Additionally, collaboration with medical professionals for rigorous clinical validation studies is essential to ensure the reliability and efficacy of the proposed models in real-world scenarios. Advancements in explainable AI methodologies can provide more detailed insights into the decision-making process, fostering trust among healthcare practitioners. Exploring patient-specific predictions could contribute to personalized medicine approaches, tailoring diagnostic insights to individual characteristics. These endeavors collectively aim to advance the current state of ovarian cancer detection, addressing challenges and paving the way for more robust, practical, and widely applicable diagnostic solutions.

## 7. Discussion

In total, 349 samples, each of which is a CT image of ovarian cancer, are classified using a deep convolutional ensemble classifier, which provided an accuracy of 98.96%. From the same patients, the clinical parameters are measured, which are subjected to an ensemble machine learning model, and an accuracy of 92.85% is obtained. Whenever a patient is subjected to health screening/diagnosis as primary care, clinical parameters such as hormonal tests and other blood investigations are prescribed. When there are diagnostic indications of possible ovarian cancer, the patient is advised to take a CT scan of the ovary. The clinical knowledge suggests that CT scans are specific in the diagnosis of ovarian cancer. This research methodology investigates the efficacy of both clinical parameters and CT scan images independently in diagnosing the disease, along with the investigation of the discrimination potential of each of the approaches (clinical parameters and CT images), which ultimately proves logically that CT scan images can not only provide a more specific but also efficient diagnosis of ovarian cancer than clinical parameters. It is advised as primary care to investigate clinical parameters to rule out conditions other than ovarian cancer. The primary contribution of this research is the evaluation and comparison of the efficacy of both clinical and CT scan images in diagnosing ovarian cancer and the justification of why CT image-based diagnosis of ovarian cancer is to be considered.

## Figures and Tables

**Figure 1 diagnostics-14-00543-f001:**
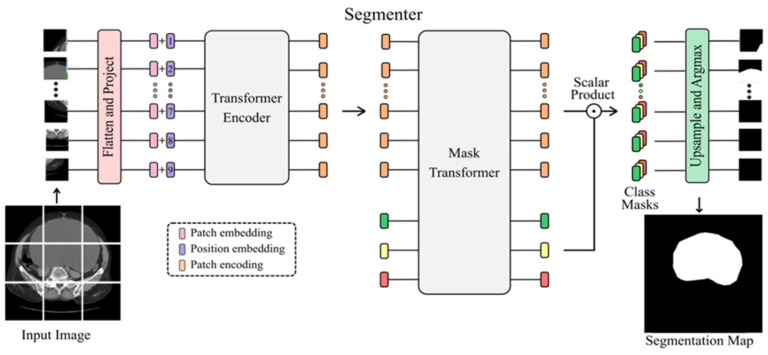
Transformer model architecture used for segmentation of benign and malignant tumors.

**Figure 2 diagnostics-14-00543-f002:**
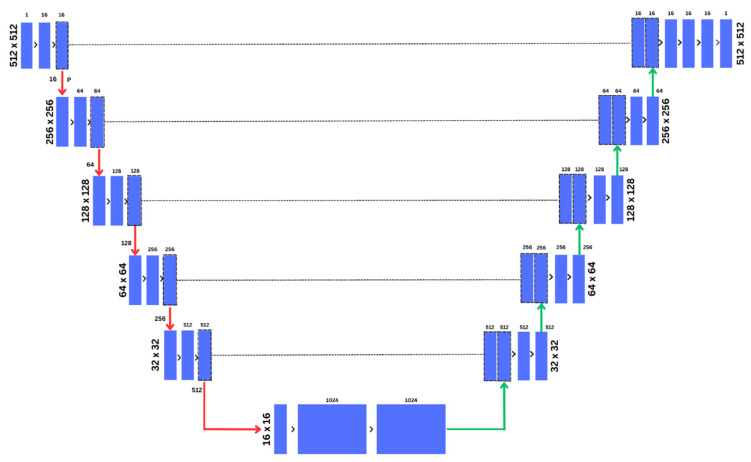
U-Net model architecture for segmentation of benign and malignant tumors.

**Figure 3 diagnostics-14-00543-f003:**
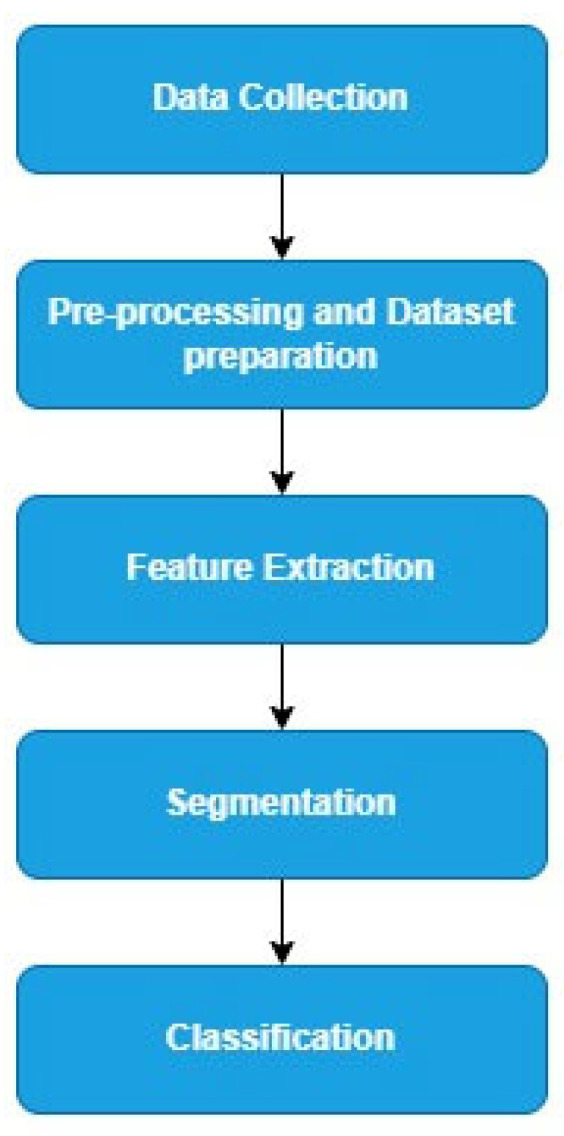
Block diagram.

**Figure 4 diagnostics-14-00543-f004:**
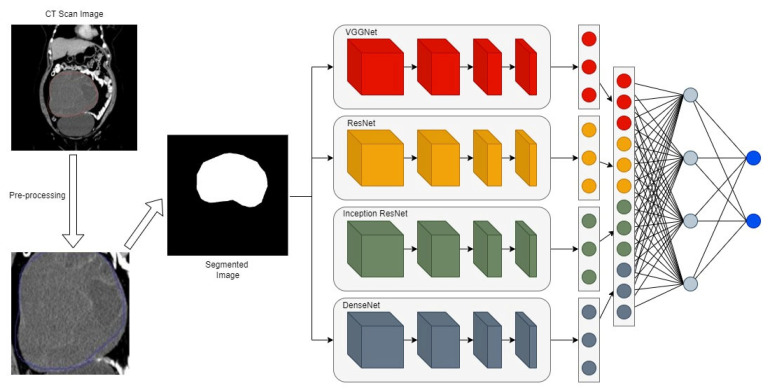
The proposed ensemble network with a four-path CNN of VGGNet, ResNet, Inception, and DenseNet.

**Figure 5 diagnostics-14-00543-f005:**
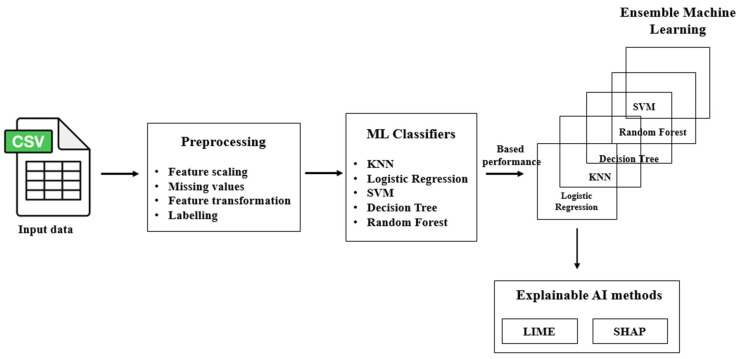
Overview of the proposed ensemble machine learning model.

**Figure 6 diagnostics-14-00543-f006:**
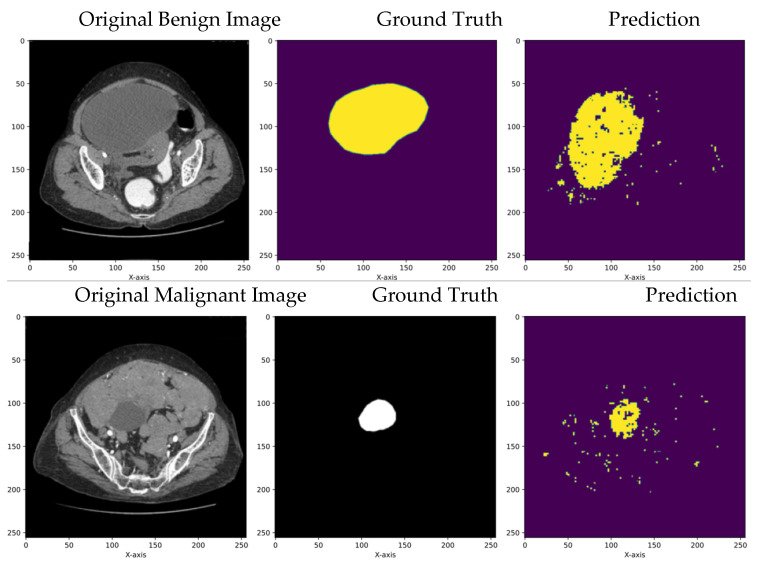
Segmentation results of benign and malignant images using UNet.

**Figure 7 diagnostics-14-00543-f007:**
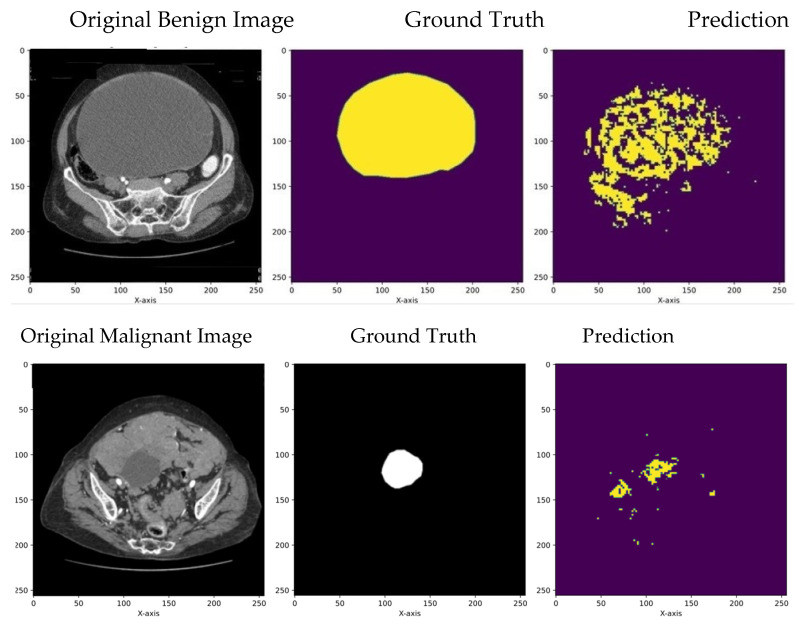
Segmentation results of benign images and malignant images using transformers.

**Figure 8 diagnostics-14-00543-f008:**
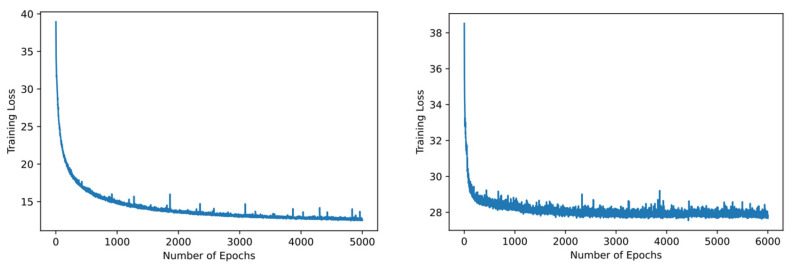
Epochs vs. loss analysis of UNet (**left**) and transformers (**right**).

**Figure 9 diagnostics-14-00543-f009:**
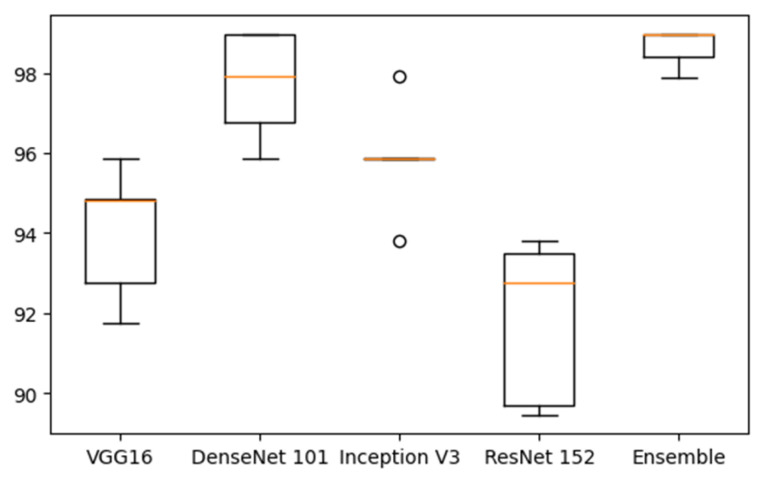
Performance of the single classifiers and the proposed ensemble model.

**Figure 10 diagnostics-14-00543-f010:**
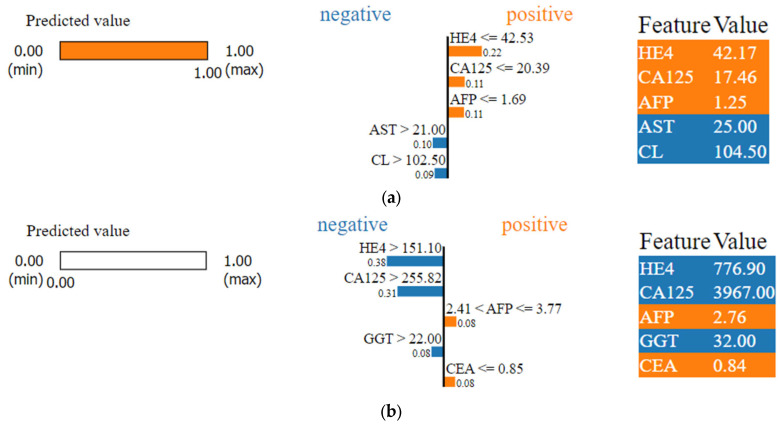
(**a**): Interpretation of the results using LIME for sample 1 and 10 (**b**) for sample 2.

**Figure 11 diagnostics-14-00543-f011:**
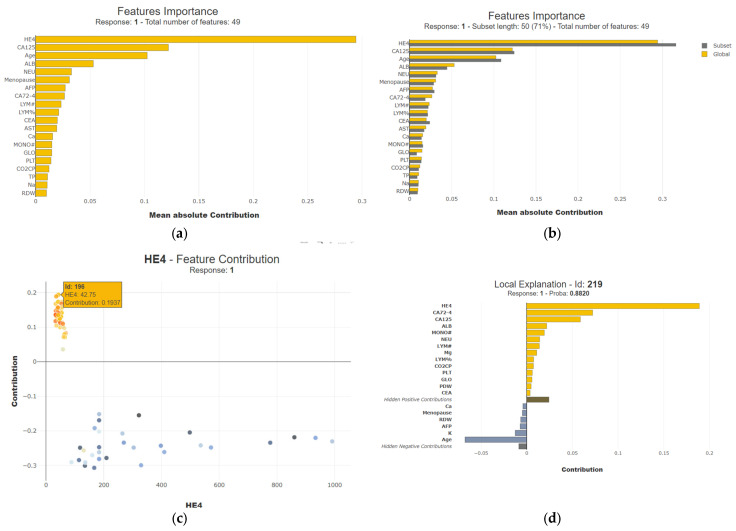
Interpretation of the results using SHAP for one sample (**a**,**c**) and impact of HE4 on the test samples (**b**,**d**).

**Table 1 diagnostics-14-00543-t001:** CNN variant specifications.

Layer	VGG16	DenseNet 101	Inception V3	ResNet 152
Size of Layers	41	101	48	152
Input image size	512 × 512 pixel	224 × 224 pixel	299 × 299 pixel	224 × 224 pixel
Convolutional Layer	13	128	42	51
Filter Size	64 & 128	3	1, 3, 5	1, 3
ReLU	5	2	42 (equivalent)	51 (equivalent)
Max Pooling	5	0	2	0
Fully Connected Layers	3	0	0	0
Softmax	1	1	1	1

**Table 2 diagnostics-14-00543-t002:** The configuration and environment settings for the experiment.

Parameters	Values
Image size for the experiment	512 × 512 pixels
Batch size	64
Number of epochs	1000
Number of hidden neurons for the ReLU	256
GPU	Nvidia RTX 3060 1.78 GHz with 3585 cores
RAM	32 GB

**Table 3 diagnostics-14-00543-t003:** Quantitative comparison of Dice and Jaccard score of the UNet model and transformers.

	UNet [28]				Transformers [27]			
	Benign		Malignant		Benign		Malignant	
	Dice	Jaccard	Dice	Jaccard	Dice	Jaccard	Dice	Jaccard
Mean	0.91	0.84	0.94	0.9	0.98	0.97	0.99	0.98
Std. Dev	0.04	0.06	0.05	0.09	0.01	0.02	0.01	0.01

**Table 4 diagnostics-14-00543-t004:** Mean accuracy of 4 individual classifiers using transfer learning.

Model	VGG16 [29]	DenseNet 101 [30]	Inception V3 [31]	ResNet 152 [32]
Mean Accuracy	94.01 ± 1.688	97.7 ± 1.362	95.87 ± 1.456	91.84 ± 2.10

**Table 5 diagnostics-14-00543-t005:** Performance of the 4-stage ensemble deep neural network.

Model	Accuracy	Precision	F1 Score
4-Stage Ensemble Deep CNN Model	98.96 ± 1.269	97.44 ± 1.2	98.7 ± 1.423

**Table 6 diagnostics-14-00543-t006:** Performance of the single classifiers.

Model	Logistic Regression	KNN	SVM	Decision Tree	Random Forest
Accuracy	90%	81.4%	82.42%	82.85%	80.57%

**Table 7 diagnostics-14-00543-t007:** Hyperparameter tuning.

Model	Manual	Randomized SearchCV	Grid SearchCV
SVM	87.4%	91.42%	90.11%
Random Forest	82.7%	88.57%	87.46%

**Table 8 diagnostics-14-00543-t008:** Performance of the ensemble model.

Model	Accuracy	Precision	F1 Score
5-Stage Ensemble Machine Learning Model	92.85%	97%	94%

## Data Availability

The dataset is available on request.

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
