# Peer review of "An Empirical Evaluation of a Novel Ensemble Deep Neural Network Model and Explainable AI for Accurate Segmentation and Classification of Ovarian Tumors Using CT Images"

_diagnostics, 2024, doi:10.3390/diagnostics14050543_

Round 1

Reviewer 1 Report

Comments and Suggestions for Authors

The authors propose an ensemble deep neural network model for detecting Ovarian cancer with CT scan images and biomarker data from all patients.

I want to convey a few important points in the article to the authors.

1. Why was the segmentation and then classification process used instead of processing and classifying the image in its raw form?

2. Data augmenting technique is applied to images. Was the biomarker data of the resulting image also used in augmentation?

Are the biomarkers the same in the new images you create from an image?

3. From which layers of the CNN models used in the ensemble model did you extract features? How many total features of an image were obtained?

4. Ensemble model architecture is more complex than known architectures. This is a disadvantage. It should be discussed why the Ensemble model was preferred.

5. How do you evaluate the classification results of image classification and biomarker data together? This situation is not understood in the article. Add it to the discussion section.

Author Response

Q1. Why was the segmentation and then classification process used instead of processing and classifying the image in its raw form?

Response:

The authors are extremely thankful to the reviewer for going through the manuscript carefully and giving valuable comments.

The reason why the segmentation was performed 1st and then classification was

  1. Since the raw images contain a large amount of information, including irrelevant details not the lesion of interest. Therefore, segmentation was applied 1st so that the model can focus only on the lesion of interest thus reducing the complexity of the classification model.
  2. Instead of considering every pixel for classification, features such as shape can be extracted from the segmentation model thus providing more discriminative information for classification. This benefits the feature extraction for the classification.
  3. Since the segmented region is given to the classifier, the classification model can be tailored to the characteristics of those regions, leading to improved accuracy.
  4. Computational efficiency as we are processing only the segmented images rather than the entire image.

The same is included in the section 3.2.1 and is highlighted in yellow.

Q2. Data augmenting technique is applied to images. Was the biomarker data of the resulting image also used in augmentation?

Are the biomarkers the same in the new images you create from an image?

Response:

Yes, the data augmentation technique was applied to increase the diversity of the training dataset. These techniques include rotation, flipping, and scaling.

Biomarkers are the ground truth provided by the radiologist. After the manual data augmentation was performed, on the newly created images the biomarkers were applied by seeing the original image marker. The newly annotated images were cross-verified by the radiologist before the segmentation algorithm was applied.

Q3. From which layers of the CNN models used in the ensemble model did you extract features? How many total features of an image were obtained?

Response:  The results obtained from each of the individual CNN models were combined and created the metadata dataset (results of all the classifiers). The Random forest classifier was run on the dataset to classify it as benign or malignant.

Q4. Ensemble model architecture is more complex than known architectures. This is a disadvantage. It should be discussed why the Ensemble model was preferred.

Response: Despite the complexity of the ensemble model architectures than the individual models, the ensemble model is used for the following reasons:

  1. The ensemble model often leads to better generalization when compared to the individual models. By combining the predictions from multiple models, the ensemble model avoids overfitting and thus captures the diverse patterns in the data.
  2. By combining the predictions, the ensemble model leverages the strength of each model, thus reducing the impact of the single model decision.
  3. Since it is a medical application, small gains in accuracy lead to significant practical improvement.

From the results, it is observed that compared to the single classifiers, ensemble architecture has improved the results.

Q5.  How do you evaluate the classification results of image classification and biomarker data together? This situation is not understood in the article. Add it to the discussion section.

Response: The details are in the discussion section and highlighted in yellow.

Reviewer 2 Report

Comments and Suggestions for Authors

The study focused on improving the early diagnosis of ovarian cancer through the development of an advanced computational model that leverages both CT scan images and biomarker data.

1. Introduction:

The statement "One in every eight women worldwide is affected by ovarian cancer" could be misleading without specifying whether this refers to lifetime risk, current incidence rates, or another metric. The lack of precise epidemiological data or a citation for this statistic may detract from the credibility of the introduction.

Several claims, such as the importance of early detection, the role of CT scans in diagnosis, and the complexity of interpreting CT images, are made without direct citations.

The introduction briefly mentions the advantages of deep learning and transformers over traditional methods but does not sufficiently acknowledge the challenges and limitations associated with these technologies.

While the introduction highlights the promise of transformers over CNNs, it lacks a nuanced discussion of contexts where CNNs might still be preferred or how these technologies can be complementary.

When mentioning "Implemented transformer models for semantic segmentation," it would be helpful to briefly specify what makes this implementation novel or superior to existing approaches.

2. Literature Review:

The literature review reads more like a list of advancements and findings without critically analyzing or synthesizing the results. It fails to discuss the limitations of the cited studies, potential conflicts in findings, or areas where results might not be directly comparable.

The review inconsistently presents the findings from various studies. It would benefit from a more structured approach.

While the focus on transformers is understandable given their emerging significance, the review could benefit from a more balanced discussion that includes cases where CNNs might still offer advantages or where hybrid models outperform pure transformer or CNN approaches.

The review does not address how the performance of transformers and CNNs might vary based on the type and size of medical imaging datasets, nor does it discuss the applicability of these models across different medical imaging modalities or contexts.

3. Methodology:

More detail on the specific configurations of the models, such as the size of the patch embeddings for the transformer or the depth and type of layers in the CNNs, would provide a clearer understanding of the methodology.

While preprocessing is mentioned, specific techniques used (e.g., normalization methods, specific data augmentation techniques) are not detailed.

The methodology assumes the superiority of certain models (e.g., transformer models for segmentation, ensemble deep learning models for classification) without providing initial evidence or rationale for these choices based on the dataset characteristics or task requirements.

While transfer learning is mentioned as a method for feature extraction, the specifics of how models are adapted or fine-tuned for the task at hand are not discussed.

The section mentions hyperparameter tuning as key to superior performance but does not detail the approach taken to tune the models. Descriptions of the search space, tuning algorithms (e.g., grid search, random search, Bayesian optimization), and validation methods would enhance the methodology's rigor.

5. Results & Discussion:

The dataset description indicates an imbalance between the number of benign and malignant images. However, there's no mention of how this imbalance was addressed during training.

While data augmentation techniques are listed, there's no indication of how these techniques were applied or their impact on the model's performance.

The experimental setup does not mention the use of cross-validation or other techniques to ensure the generalizability of the model across different data subsets.

The use of LIME and SHAP explainer models is commendable for model interpretability. However, the discussion on the insights gained from these models is somewhat superficial. A deeper analysis of how these explanations influenced model refinement or clinical understanding would be valuable.

Author Response

Q1. 1. Introduction:

The statement "One in every eight women worldwide is affected by ovarian cancer" could be misleading without specifying whether this refers to lifetime risk, current incidence rates, or another metric. The lack of precise epidemiological data or a citation for this statistic may detract from the credibility of the introduction.

Several claims, such as the importance of early detection, the role of CT scans in diagnosis, and the complexity of interpreting CT images, are made without direct citations.

The introduction briefly mentions the advantages of deep learning and transformers over traditional methods but does not sufficiently acknowledge the challenges and limitations associated with these technologies.

While the introduction highlights the promise of transformers over CNNs, it lacks a nuanced discussion of contexts where CNNs might still be preferred or how these technologies can be complementary.

When mentioning "Implemented transformer models for semantic segmentation," it would be helpful to briefly specify what makes this implementation novel or superior to existing approaches.

Response:

The authors are extremely thankful to the reviewer for going through the manuscript carefully and giving valuable comments.

The entire Introduction section is rewritten by incorporating the reviewer's comments. The responses are highlighted in the Introduction section in yellow.

Q2. 2. Literature Review:

The literature review reads more like a list of advancements and findings without critically analyzing or synthesizing the results. It fails to discuss the limitations of the cited studies, potential conflicts in findings, or areas where results might not be directly comparable.

The review inconsistently presents the findings from various studies. It would benefit from a more structured approach.

While the focus on transformers is understandable given their emerging significance, the review could benefit from a more balanced discussion that includes cases where CNNs might still offer advantages or where hybrid models outperform pure transformer or CNN approaches.

The review does not address how the performance of transformers and CNNs might vary based on the type and size of medical imaging datasets, nor does it discuss the applicability of these models across different medical imaging modalities or contexts.

Response:

We are thankful to the reviewer for suggesting the changes and making the paper more technical and strong. All the questions related to the literature review are incorporated and the responses are highlighted in the Related Work section of the paper. 

Q3. 3. Methodology:

3.1 More detail on the specific configurations of the models, such as the size of the patch embeddings for the transformer or the depth and type of layers in the CNNs, would provide a clearer understanding of the methodology.

3.2 While preprocessing is mentioned, specific techniques used (e.g., normalization methods, specific data augmentation techniques) are not detailed.

3.3 The methodology assumes the superiority of certain models (e.g., transformer models for segmentation, ensemble deep learning models for classification) without providing initial evidence or rationale for these choices based on the dataset characteristics or task requirements.

3.4 While transfer learning is mentioned as a method for feature extraction, the specifics of how models are adapted or fine-tuned for the task at hand are not discussed.

3.5 The section mentions hyperparameter tuning as key to superior performance but does not detail the approach taken to tune the models. Descriptions of the search space, tuning algorithms (e.g., grid search, random search, Bayesian optimization), and validation methods would enhance the methodology's rigor.

Response:  

3.1 Configuration of the Transformer model is provided in section 3.1.1 and the same is highlighted in yellow.

3.2 & 3.3 Choices of the model & Preprocessing details are mentioned and highlighted in yellow in section 3.2.3.

3.4 Transfer learning model details are provided in Table 1 and the same is highlighted in section 3.2.3

3.5 Hyperparameter tuning details are recorded in Table 7 in section 5.3 and the same is highlighted in yellow.

Q4. Results & Discussion:

4.1 The dataset description indicates an imbalance between the number of benign and malignant images. However, there's no mention of how this imbalance was addressed during training.

4.2 While data augmentation techniques are listed, there's no indication of how these techniques were applied or their impact on the model's performance.

4.3 The use of LIME and SHAP explainer models is commendable for model interpretability. However, the discussion on the insights gained from these models is somewhat superficial. A deeper analysis of how these explanations influenced model refinement or clinical understanding would be valuable.

Response:

4.1. Since the data augmentation techniques were used, the number of samples was increased during the training phase. The details are updated in the paper and the same is highlighted in section 4.2.1

4.2. Data augmentation techniques used are highlighted in the section 4.2.1

4.3. The detailed interpretation of LIME and SHAP explainers is provided and the same is highlighted in sections 5.4.1 and 5.4.2

Round 2

Reviewer 2 Report

Comments and Suggestions for Authors The manuscript has been sufficiently improved to warrant publication in Diagnostics